# Analysis of the Territorial Vocalization of the Pheasants *Phasianus colchicus*

**DOI:** 10.3390/ani12223209

**Published:** 2022-11-19

**Authors:** Piotr Czyżowski, Sławomir Beeger, Mariusz Wójcik, Dorota Jarmoszczuk, Mirosław Karpiński, Marian Flis

**Affiliations:** Department of Animal Ethology and Wildlife Management, University of Life Sciences in Lublin, Akademicka 13, 20-950 Lublin, Poland

**Keywords:** common pheasant, *Phasianus colchicus*, vocalization, behavior, mating season

## Abstract

**Simple Summary:**

Research on bird vocalizations is a useful tool for further research on behavior. Singing behavior to reproductive behavior such as territory formation and mate choice. The aim of the study was to assess the impact of the duration of the mating season and the time of day on the parameters of the vocalization of pheasants *Phasianus colchicus* (duration of vocalization, frequency of the sound wave, intervals between vocalizations). The study consisted of analyses of recordings of the sounds of crowing pheasant cocks inhabiting an area in Lublin. In the study, pheasant vocalizations that were recorded in the morning (6^00^–8^00^) and in the afternoon (16^00^–18^00^) between April and June 2020 were analyzed. Statistically significant differences in the distributions of the values of all variables between the analyzed months were demonstrated. The duration of vocalization was significantly shorter in the morning, which indicates that the cooks are more active at this time of day in the study area. The time of day was also shown to have an impact on the peak amplitude frequencies, which had the highest values in the morning. The results of the present study conducted in urban areas can be the basis for a comparison with populations of pheasants inhabiting agricultural areas.

**Abstract:**

The aim of the study was to assess the impact of the duration of the mating season and the time of day on the parameters of the vocalization pheasants (duration of vocalization, frequency of the sound wave, intervals between vocalizations). In the study, pheasant vocalization recorded in the morning (6^00^–8^00^) and in the afternoon (16^00^–18^00^) between April and June 2020 was analyzed. In total, the research material consisted of 258 separate vocalizations. After recognition of the individual songs of each bird, frequency-time indicators were collected from the samples to perform statistical analysis of the recorded sounds. The duration of the first syllable [s], the duration of the second syllable [s], the duration of the pause between the syllables [s], the intervals between successive vocalizations [min], and the peak frequency of the syllables I and II [Hz] were specified for each song. The duration of the syllables and the pauses between the syllables and vocalizations were determined through evaluation of spectrograms. The peak amplitude frequencies of the syllables were determined via time-frequency STFT analysis. Statistically significant differences in the distributions of the values of all variables between the analyzed months were demonstrated. The longest duration of total vocalization and the shortest time between vocalizations were recorded in May. Therefore, this month is characterized by the highest frequency and longest duration of vocalization, which is related to the peak of the reproductive period. The time of day was found to exert a significant effect on all variables except the duration of syllable II. The duration of vocalization was significantly shorter in the morning, which indicates that the cooks are more active at this time of day in the study area. The highest peak amplitude frequencies of both syllables were recorded in April, but they decreased in the subsequent months of observation. The time of day was also shown to have an impact on the peak amplitude frequencies, which had the highest values in the morning.

## 1. Introduction

Analyses of bird sounds are regarded as an important method for studying bird ecology and behavior [1,2,3]. Birdsong is uniquely amenable and attractive for behavioral analysis because it is a structured behavior, repeated with a high degree of stereotypy. This makes it relatively easy to detect and characterize song structure and components and to relate singing behavior to reproductive behavior [4]. There is little research on the acoustic characteristics of crowing pheasant [5,6] despite the ease of observation of the characteristic mating ritual with loud vocalization.

The common pheasant (*Phasianus colchicus* Linnaeus, 1758) belonging to the order Galiiformes, family Phasianidae, subfamily Phasianinae, and genus *Phasianus* L. is the most widespread pheasant species in the world [7,8].

At the beginning of spring, male pheasants establish, guard, and defend territories against other males through loud vocalization accompanied by stretching the body, flapping the wings, tossing the head, and lifting and fanning the tail. This phenomenon, known as crowing, is supposed to attract females and chase other males away from the territory [9]. The intensity of crowing depends on various factors, e.g., the density of birds in the area, the activity of other cooks, the presence of other sounds, etc. [5]. During the peak of the mating season, vocalizing cooks can be heard frequently, even every few minutes. In territories located close to each other, males present in the neighboring areas respond to each other by crowing. Males exhibiting greater territoriality, moving less intensively during the day, and crowing more frequently are the most attractive to females [9]. The most attractive territories are highly diverse, provide better access to food and nesting sites, and ensure easy escape from potential threats [10]. Females select crowing males probably based on the morphology [11,12], mating behavior [13], and major histocompatibility complex (MHC) genes [14]. The sound made by the cooks can also provide hens with acoustic information on the physical fitness of the male, which indirectly reflects the quality of the habitat occupied by the male [2]. The mating system of pheasants is based on the protection of females from predators and from the excessive energy expenditure of females associated with the search for other males. [15]. Thus, hens choose the partner as part of natural selection for reproduction of individuals with the desired phenotypic characteristics. On the one hand, the territorial vocalization is a signal informing other males that the area is occupied and potential rivals in the fight for breeding can be eliminated; on the other hand, it is intended to attract as many hens as possible [10,16].

Thorough knowledge of the mating ritual can contribute to elucidation of the mechanisms of preferences of female pheasants in the choice of the cock. The results of the present study conducted in urban areas can be the basis for a comparison with populations of pheasants inhabiting agricultural areas. Increasing numbers of studies indicate changes in vocalizations in various animal species induced by exposure to anthropogenic noise and differentiation of the acoustic signal between populations of the same species in urban and non-urban habitats [17,18,19].

Research on avian vocalizations provides a useful tool for monitoring populations and research on behavior. It is relatively easy to detect and characterize song structure and components and to relate singing behavior to reproductive behavior such as how it can affect territory formation and mate choice.

To date, there have been few studies on the subject of pheasant vocalization. The results of the present study conducted in urban areas can be the basis for a comparison with populations of pheasants inhabiting agricultural areas. The aim of the study was to assess the impact of the duration of the mating season and the time of day on the parameters of the vocalization (duration of vocalization, frequency of the sound wave, intervals between vocalizations).

## 2. Materials and Methods

The study consisted of analyses of recordings of the sounds of four wild crowing pheasant cocks inhabiting in urban areas (Lublin, Poland). The total length of the recordings made in the different terms ranged from 12 to 60 min. The recordings were made using a Zoom H1 portable audio recorder. The recordings were made from a distance that did not scare the birds away.

Prior to subsequent analysis and archivization, the files with the recorded cock pheasant vocalization were divided into 3-min fragments and analyzed in the Cool Edit Pro program. The pheasants studied were birds living in the wild in the city park. We recorded the voices of individual roosters separately for each individual, which was preceded by observations in the field. The males had access to the hens. The choice of days on which recordings were made was irregular and depended on the activity of the recorded roosters. The number of recordings for analysis represents a random sample of all noises. In the study, pheasant vocalizations recorded in the morning (6^00^–8^00^) and in the afternoon (16^00^–18^00^) between April and June 2020 on randomly selected days were analyzed. In total, the research material consisted of 258 separate vocalizations (Table 1).

After recognition of the individual songs of each bird, frequency-time indicators were collected from the samples to perform statistical analysis of the recorded sounds. This was carried out by analyzing the sonograms of individual pheasant vocalizations using a sound analysis program. The duration of the first syllable [s], the duration of the second syllable [s], the duration of the pause between the syllables [s], the intervals between successive vocalizations [min], and the peak frequency of the syllables I and II [Hz] were specified for each song. The duration of the syllables and the pauses between the syllables and vocalizations were determined through evaluation of spectrograms (Figure 1). The peak amplitude frequencies of the syllables were determined via time-frequency STFT analysis.

The statistical analysis of the results was performed with the use of the Statistica 13.3 PL package. Since the distribution of the analyzed traits deviated from normality significantly, non-parametric (rank) tests were used to analyze the significance of the differences between the distributions. The Mann-Whitney *U* rank test (Z statistics) was used to compare two groups (both times of day). In turn, three groups (months) were compared with the non-parametric Kruskal-Wallis ANOVA and the multiple pairwise comparison of mean ranks.

When significant differences were found by the comparison of many groups, the result of the multiple comparison tests was marked with letters on boxplots. Groups marked with the same letter did not differ significantly (the same homogeneous group), whereas significant differences were found between groups marked with different letters.

The positional average measure, i.e., the median, and quartiles were used to describe the distributions. The distribution of the analyzed features of the selected parameters in the individual months is illustrated in categorized boxplots based on positional measures (median, quartiles). The normality of the distributions of the traits was assessed using the Shapiro-Wilk test. The results were statistically significant at the typical significance level of *p* < 0.05.

A factor analysis was performed to investigate the structure of the relationships between the variables and reduce their number. The number of factors required for further analysis was determined using the Kaiser criterion and the cumulative percentage of explained variance (>80%). The maximization of the variance of factor loadings was performed using the Varimax rotation.

## 3. Results

The total vocalization consisted of two separate syllables separated by a pause; the first syllable was longer than the second one (Figure 2). The mean duration of the second syllable of all analyzed sounds was 0.14 [s], i.e., it was significantly shorter than the duration of the first syllable of 0.22 [s] and the pause between the syllables of 0.22 [s] (Kruskal-Wallis ANOVA: χ^2^ = 298,397; *p* < 0.001) (Figure 3). The mean duration of the entire vocalization was 0.58 [s].

## 4. Distribution of Variables According to the Month

Since all the variables compiled according to the month of vocalization deviated from the normal distribution significantly and showed heterogeneity of variance, the positional average measure, i.e., the median, was used for the description of the variables in addition to the mean values (Table 2). The results of the non-parametric Kruskal-Wallis analysis of variance revealed significant differences in the distributions of all variables in the months when the vocalization was recorded except for the duration of the pause between consecutive vocalizations, where the effect of the vocalization recording month on the differences in this variable was close to significance. The shortest pause between the vocalizations was recorded in May; hence, the higher frequency of pheasant vocalizations in this month. The month of observation exerted a clear effect on the total vocalization time, as its mean values in May were significantly higher than in April and June (Table 2). It was also shown that the vocalization month had an impact on the individual components of vocalization. The longest duration of syllable I was recorded in May, and this value was statistically significantly different from that recorded in June. In turn, the pause between the syllables was significantly longer in May than that recorded in April only. The duration of syllable II was significantly longer in April than in June. The comparison of the distribution of the peak amplitude frequency of syllable I indicated that the value of this variable was higher in April than in May and June. Similarly, the highest peak amplitude frequency of syllable II was recorded in April. It was significantly different from that recorded in May.

## 5. Distribution of Variables According to the Time of Day

The comparison of the distribution of the variables depending on the time of day revealed statistical differences in all cases, except for the duration of syllable II (Table 3). With regard to the time of day, the duration of the pause between consecutive vocalizations was significantly shorter in the morning than in the afternoon, which indicates a higher rate of vocalizations in the morning. The total vocalization was significantly shorter in the morning, similar to the duration of syllable I and the pause between the syllables. The comparison of the distribution of the frequency of the vocalization syllables showed significantly higher peak amplitude frequencies of syllables I and II in the morning.

## 6. Factor Analysis

As indicated by the correlation coefficients, some variables were strongly correlated, i.e., the duration of total vocalization with the duration of syllable I (r = 0.5909), the duration of the pause between the syllables (r = 0.4446), and the duration of syllable II (r = 0.6476). This seems obvious since these are components of the duration of the total vocalization. A similar correlation was exhibited by the frequencies of syllables I and II (r = 0.6136).

The use of Varimax rotation allowed for a better representation of the variables observed in the factor space (Table 4). As shown in the table, the first, second, third, and fourth factors explained 28%, 24%, 18%, and 14% of the variance, respectively.

The first factor was the vocalization time, with loadings of this factor directly related to the duration of vocalization. The second factor was the frequency of vocalization with the peak amplitude frequencies of both syllables as the variables. The third factor was the pause between the syllables; although it was part of the duration of vocalization and strongly correlates with its other components, the factor analysis indicated it as a separate factor. The fourth factor, i.e., the pause between vocalizations, did not correlate with any other primary variable (it was not related to any other factor).

The two-dimensional plot of the factor space projection onto the plane of the first two factors (Figure 4) shows distinct clusters of variables constituting the first factor (vocalization time) and the second factor (vocalization frequency).

The plot of the factor space projection onto the plane of the third and fourth factors (Figure 5) shows a clear separation of the variables constituting the third factor (pause between syllables) and the fourth factor (pause between vocalizations).

## 7. Discussion

The present study demonstrated the highest frequency and the longest duration of vocalization of the pheasants in May, which is associated with the peak of the reproductive season [20]. Importantly, the pause between subsequent vocalizations and the frequency of vocal responses are strongly influenced by other factors, e.g., the density of birds, the activity of other cocks, the presence of other sounds, etc. [5], which has also been confirmed in studies on songbird species [21]. As reported by Luukkonen et al. [22], age, weather variables, and subspecies has no effect on the frequency of the vocal response in pheasants.

In this study, the sound frequency range was between 100 and 4500 Hz, and 50% of all observations (between quartiles I and II) were in the range of 1000–1010 Hz.

In the case of pheasants, the frequency audibility is in the range of 250–10500 Hz, and the most pronounced frequency of the syllables of the crowing pheasant’s song is usually in the range from 800 Hz to 1000 Hz [5,6]. Some authors [23,24] emphasize the variability of the vocalization sound frequency in birds living in habitats transformed by human activity. There is also a relationship between the frequency of vocalizations and the weight of gonads, and thus the level of testosterone [after 5], which undergoes cyclical changes [25]. Therefore, it is assumed that hormonal factors may have a significant impact on the vocalization behavior, and the analysis of sounds produced by birds may be an indirect determinant of the individual quality of cocks. Studies on the impact of the environment on the sound frequency of bird songs [21] have demonstrated that birds can use variations in the frequency range to improve sound propagation in different environments.

A three-year study with spectrographic analysis [5] of the structure of vocal responses of pheasants showed that the inter-individual variability of vocalization is greater than the intra-individual variability; hence, it is possible to distinguish individuals through detailed spectrographic analysis. Investigations of the acoustic structure in partridges Perdix perdix [26] have shown considerable variability of the vocalization structure depending on the season of the year.

The factor analysis showed no correlations of the duration of the pause between syllables with the duration of the syllables and the duration of the total vocalization. This finding may indicate that the duration of the pause between syllables is a separate element of vocalization and can be regarded as a separate indicator in the analysis of vocalization in pheasants. This is in line with the Temporal Method, which takes account of the pauses in the song stream [27]. A single vocalization in pheasants consists of one pause between syllables. In the present study, in 50% of all observations (between quartiles I and II), it was in the range of 0.18–0.28 s. The differences in the description of vocalizations between individuals were not analyzed in this study, but the length of the pause between syllables may be a quality-reflecting trait of individuals, as reported by Laje et al. [28].

The present study was conducted in urban areas (Lublin, Poland). As shown by Slabbekoorn and Boer-Visser [29], the growing urbanization and the continuous increase in noise levels in cities have a negative effect on the reproductive behavior of many bird species, disturbing, e.g., the territorial structure of species that establish their territories through vocalization. Undoubtedly, investigations of the behavior of species inhabiting urban areas largely contribute to understanding behavioral changes in species living in cities.

## 8. Conclusions

Statistically significant differences in the distributions of the values of all variables between the analyzed months were demonstrated, except for the time between vocalizations, where the difference was close to significance. The longest duration of total vocalization and the shortest time between vocalizations were recorded in May. Therefore, this month is characterized by the highest frequency and longest duration of vocalization, which is related to the peak of the reproductive period.

The time of day was found to exert a significant effect on all variables except the duration of syllable II. The duration of vocalization was significantly shorter in the morning, which indicates that the cooks are more active at this time of day in the study area.

The highest peak amplitude frequencies of both syllables were recorded in April, but they decreased in the subsequent months of observation. The time of day was also shown to have an impact on the peak amplitude frequencies, which had the highest values in the morning.

The factor analysis distinguished four factors: the vocalization time, the vocalization frequency, the pause between syllables, and the pause between vocalizations.

The results of the present study conducted in urban areas can be the basis for a comparison with populations of pheasants inhabiting agricultural areas.

## Figures and Tables

**Figure 1 animals-12-03209-f001:**
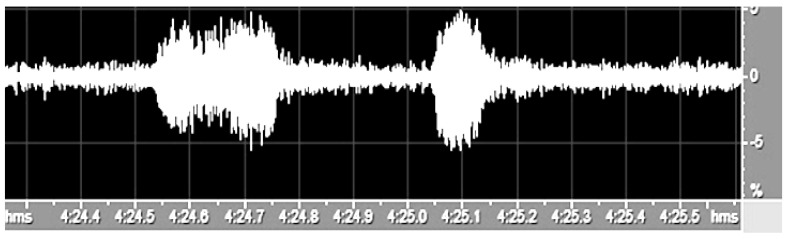
Screenshot of a cook pheasant vocalization oscillogram in Cool Edit Pro.

**Figure 2 animals-12-03209-f002:**
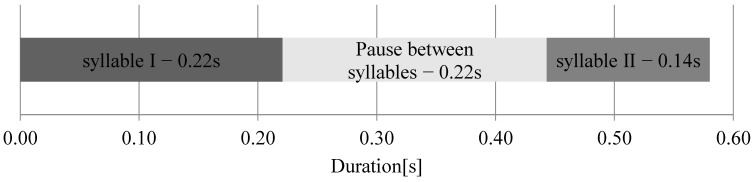
The mean duration of vocalization.

**Figure 3 animals-12-03209-f003:**
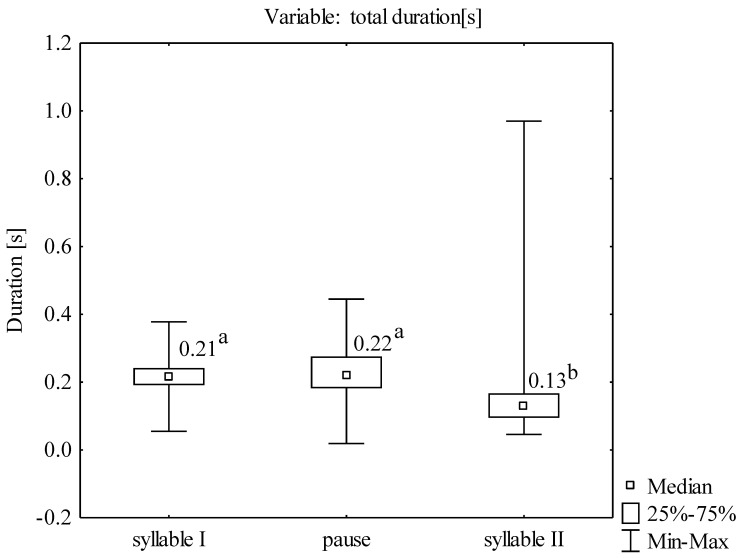
The distribution of the duration of individual elements of vocalization. The variables marked with different letters differ significantly at *p* < 0.05; (Kruskal-Wallis ANOVA, multiple comparison test).

**Figure 4 animals-12-03209-f004:**
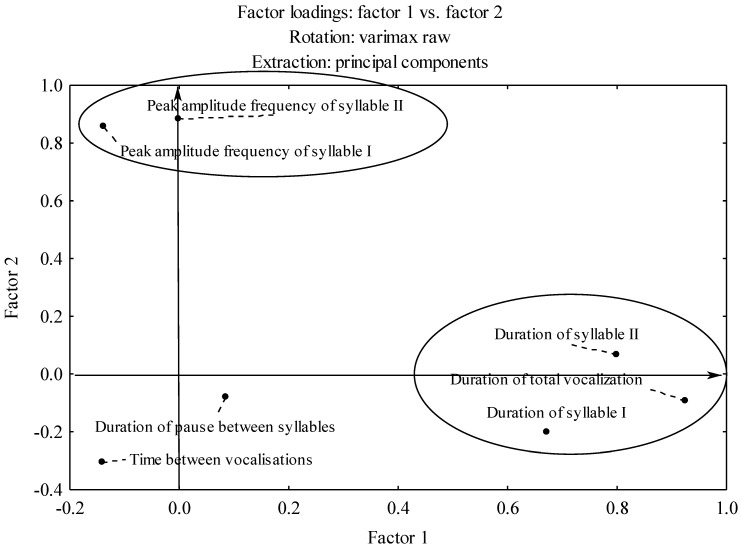
The plot of the projection of variables onto the plane of the first two factors after Varimax rotation.

**Figure 5 animals-12-03209-f005:**
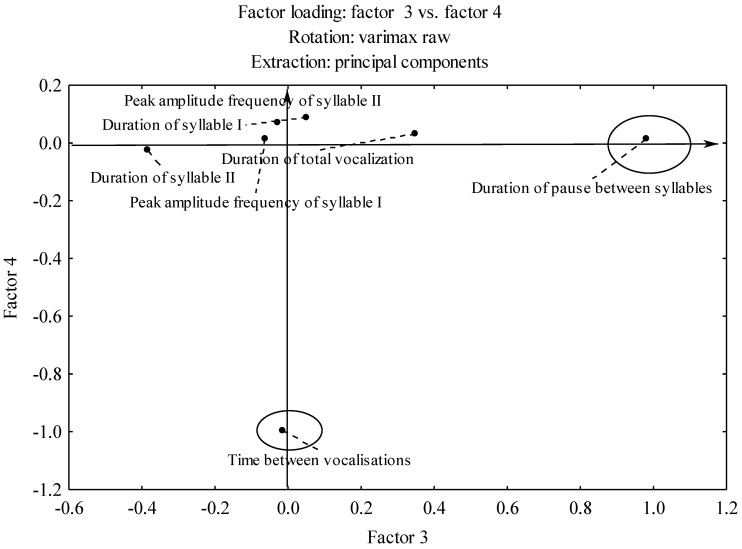
The plot of the projection of the variables onto the planes of the third and fourth factors after Varimax rotation.

**Table 1 animals-12-03209-t001:** The number of vocalizations in the different months and time of day.

No. of Cock	Month	Time of Day: Morning	Time of Day: Afternoon	Total in Rows
1	April	11	25	36
1	May	3	25	28
1	June	8	6	14
Total		22	56	78
2	April	18	8	26
2	May	6	11	17
2	June	12	6	18
Total		36	25	61
3	April	14	5	19
3	May	12	12	24
3	June	7	3	10
Total		33	20	53
4	April	23	4	27
4	May	9	15	24
4	June	13	2	15
Total		45	21	66
Total in columns		136	122	258

**Table 2 animals-12-03209-t002:** The distribution of variables according to the month.

Variables	April (*n* = 108)	May (*n* = 93)	June (*n* = 57)	Chi^2^	*p*-Value
Time between vocalizations [min]	3.0 (2.0–4.0)	2.5 (1.5–3.5)	3.0 (2.0–5.0)	5.8258	0.0543
Duration of total vocalization [s]	0.549 ^a^ (0.515–0.649)	0.606 ^b^ (0.558–0.639)	0.571 ^a^ (0.510–0.608)	13.6055	0.0011
Duration of syllable I [s]	0.206 ^ab^ (0.188–0.251)	0.228 ^a^ (0.198–0.251)	0.207 ^b^ (0.188–0.227)	9.3476	0.0093
Duration of pause between syllables [s]	0.205 ^a^ (0.149–0.262)	0.222 ^b^ (0.195–0.286)	0.234 ^ab^ (0.190–0.274)	11.6767	0.0029
Duration of syllable II [s]	0.149 ^a^ (0.095–0.168)	0.125 ^ab^ (0.103–0.169)	0.107 ^b^ (0.091–0.139)	8.2507	0.0162
Peak amplitude frequency of syllable I [Hz]	1065 ^a^ (1043–1135)	1058 ^b^ (1012–1085)	1041 ^b^ (1030–1057)	19.3389	0.0001
Peak amplitude frequency of syllable II [Hz]	1059 ^a^ (1036–1131)	1036 ^b^ (1014–1069)	1046 ^ab^ (1026–1069)	15.5683	0.0004

^ab^ The variables marked with different letters differ significantly at *p* < 0.05; (Kruskal-Wallis ANOVA, multiple comparison test).

**Table 3 animals-12-03209-t003:** The distribution of variables according to the time of day.

Variables	Morning (*n* = 136)	Afternoon (*n* = 122)	Z	*p*-Value
Time between vocalizations [min]	2.0 (2.0–3.0)	3.0 (2.0–4.0)	−3.8113	0.0001
Duration of total vocalization [s]	0.548 (0.512–0.605)	0.608 (0.547–0.649)	−4.5554	0.0001
Duration of syllable I [s]	0.204 (0.188–0.231)	0.231 (0.198–0.255)	−4.3176	0.0001
Duration of pause between syllables [s]	0.201 (0.163–0.272)	0.246 (0.200–0.280)	−3.5369	0.0004
Duration of syllable II [s]	0.141 (0.092–0.174)	0.121 (0.100–0.158)	1.5358	0.1246
Peak amplitude frequency of syllable I [Hz]	1063 (1037–1135)	1050 (1018–1071)	3.4115	0.0006
Peak amplitude frequency of syllable II [Hz]	1052 (1028–1132)	1046 (1016–1062)	3.2151	0.0013

**Table 4 animals-12-03209-t004:** The factor loadings.

Variables	Factor Loadings (Varimax Raw) Extraction: Principal (The Marked Loads Are >0.600)			
	Factor I—Vocalization Time	Factor II—Frequency of Vocalizations	Factor III—Pause between Syllables	Factor IV—Pause between Vocalizations
Time between vocalizations [min]	−0.026482	−0.050769	−0.015659	−0.995233
Duration of total vocalization [s]	**0.928214**	−0.084018	0.348651	0.033440
Duration of syllable I [s]	**0.660839**	−0.222706	0.051595	0.086672
Duration of pause between syllables [s]	0.089415	−0.049110	**0.980179**	0.015422
Duration of syllable II [s]	**0.806466**	0.068687	−0.384649	−0.022500
Peak amplitude frequency of syllable I [Hz]	−0.121584	**0.880566**	−0.064295	0.012807
Peak amplitude frequency of syllable II [Hz]	0.008652	**0.896802**	−0.028492	0.071599
Expl. Var.	1.972230	1.646015	1.238115	1.005153
Prp. Totl.	0.281747	0.235145	0.176874	0.143593

## Data Availability

The data presented in this study are available on request from the Marian Flis.

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
