# Peer review of "Analysis of the Territorial Vocalization of the Pheasants Phasianus colchicus"

_animals, 2022, doi:10.3390/ani12223209_

Round 1

Reviewer 1 Report

The article is well conceptualised and statistically developed, but it is too descriptive.

I think the Discussion could elaborate on some of the aspects:

- Since it has been shown that the sound frequency of birdsong can be varied to enhance sound propagation in different environments, how could this be applied to pheasants? Comments could be made on whether differences have been observed in pheasant songs depending on habitats (open fields or more densely vegetated areas).

- One of the possible improvements would have been to compare songs in urban and natural environments. If differences exist in other works, it may be cited in the discussion.

Author Response

Responding to the Reviewer's comments 1

Thank you for your right comments, it would be really valuable to expand our work to include other environments. Unfortunately, we could not find any research on the vocalization of pheasants in the available literature. We are aware that the literature on songbird vocalization is rich, but it is difficult to relate to our research unequivocally. Additionally, we did not want to increase the size of the article with other genres.

Reviewer 2 Report

This is an interesting study but rather limited. The variation in vocalisation frequency between  birds has been carefully documented, but the study  would have been of greater interest and relevance if it were associated with harem size or breeding success. If the authors have such data, they should certainly include them in the results. A testable hypothesis would lend more focus to the study.

The title needs to be more descriptive. For example, what is the study trying to determine.

The simple summary should also include the relevance of the results to the behavioural ecology of the birds. Similarly the abstract describes the details of the study but does not give the context and the relevance of the results to avian ecology.

The methods need to include the basis upon which these four cocks were chosen, whether they are captive or wild birds, and whether other observations have been conducted on them. How did the observers carry out the observations without disturbing the birds?

A map of the study area with the location of each bird's territory would be useful. 

Discussion can be improved by highlighting the main points, citing other sources with similar studies and improving the structure which is loose at present. The emphasis on the patterns of vocalisation is of limited interest to a general audience as it is not based on an ecological or evolutionary context. The authors have cited some papers with the same pheasant species, but the results of those studies have not been presented nor discussed in depth, i. e. for instance is there a relationship between the pattern of vocalisation and mating success?   

Author Response

Responding to the Reviewer's comments 2

We agree with the Reviewer that the work would be richer if we relate our results to the success of reproduction and the size of the harem. Unfortunately, we do not have such results at the moment, because this stage of research has already been completed. We plan to continue our research and supplement it with this information, but only after the final completion of such research will we try to publish the obtained results. As suggested by the Reviewer, in the methodology of work, we added information on the origin of the birds (wild-living specimens in the urban area) and the method of recording their sounds (directional microphone, recording distance that did not scare the birds). We are not sure if the map with the marked location of the birds would bring something new to the work at this stage of research, but would only increase the volume of the article, which we consider to be pointless. We believe that the importance of our research for ecology and practice has been sufficiently addressed: “Thorough knowledge of the mating ritual can contribute to elucidation of the mechanisms of preferences of female pheasants in the choice of the cock. The results of the present study conducted in urban areas can be the basis for a comparison with populations of pheasants inhabiting agricultural areas. Increasing numbers of studies indicate changes in vocalizations in various animal species induced by exposure to anthropogenic noise and differentiation of the acoustic signal between populations of the same species in urban and non-urban habitats ".We supplemented the methodology with elements suggested by the Reviewer. We would like to thank the reviewer for valuable tips.

Reviewer 3 Report

Dear Authors,

Thank you for submitting this paper that evaluates the song structure of ring necked pheasants. This is an interesting and unusual study.

At current however, there are serious revisions required in the manuscript to ensure the work is scientifically robust. I have attached the PDF version of the manuscript with specific comments. Additionally, please consider the following points: 

1. Make sure the Animals template and font styles is used correctly.

2. Background. The wider research in this area is poorly explained. Please provide a much clearer overview of the existing studies.

3. Method. The methods need to be provided in much clearer detail. There is poor information on where the pheasants were, whether they were captive or wild, and how the observations were conducted. This needs to be explained in a level that is repeatable.

4. Wording. Some sentences do not make sense. A full proof read is needed.

5. Implications. At current it is not clear why this study matters. Please provide a much clearer explanation of the value of this research, along with the future directions.

Author Response

Responding to the Reviewer's comments 3

We would like to thank the Reviewer for valuable tips and comments on the text. As suggested, we tried to correct all comments made in the text. Regarding the remark regarding the importance of our research for ecology and practice, we believe that it has been sufficiently addressed: “Thorough knowledge of the mating ritual can contribute to elucidation of the mechanisms of preferences of female pheasants in the choice of the cock. The results of the present study conducted in urban areas can be the basis for a comparison with populations of pheasants inhabiting agricultural areas. Increasing numbers of studies indicate changes in vocalizations in various animal species induced by exposure to anthropogenic noise and differentiation of the acoustic signal between populations of the same species in urban and non-urban habitats ". Additionally, we have supplemented the chapter conclusions. We agree that it would be valuable to expand our work to include a broader literature. Unfortunately, the available literature lacks new research on the vocalization of pheasants. We recognize that there is a wealth of literature on songbird vocalization, but we did not want to expand this article to include other species. Nevertheless, we supplemented the study with two more items, in our opinion significant for the obtained results. As suggested by the Reviewer, in the methodology of work, we added information on the origin of birds (wild-living specimens in urban areas) and the method of recording their sounds (directional microphone, from a distance that did not scare the birds away).

Addressing specific comments

  1. There needs to be some background here - why does the study matter?

In line with this suggestion, we have highlighted the relevant attention in the conclusions

  1. for what animal?

We improved.

  1. There needs to be more information on what was found in this study.

We improved.

  1. for what animal?

We improved.

  1. When first mentioning a species, please include its scientific name.

We improved.

  1. What are the implications of this study?

We believe that we have explained sufficiently in the introduction, but additionally highlighted the relevant comment in the conclusions

  1. Some of the key words are already included in the title. Remove any key words that are in the title and use new terms to increase paper discoverability

It seems to us that there are different terms in the keywords than those in the title, except for the Latin name pheasant

  1. Why? this needs to be covered in more detail.

We improved, with citation.

  1. This needs to be cited
  2. This needs to be cited.

We added one citation

  1. do you mean the same pheasant crows every 3 minutes? This is unclear.

We have used the plural here.

  1. probably is a weak term. what does the research say? What do they select for more specifically?

We did not want to elaborate on the topic, but provided a reference to the literature

  1. this doesn’t make sense. Can you rephrase?

We improved.

  1. Please check wording here.

We improved lekking on crowing.

  1. state location here.

We improved.

  1. the word rooster means a male chicken. Please rephrase here.

We improved rooster on cock pheasant

Table 1. Just include the month name.

We improved.

Table 1. This is unclear. Where were the birds? Are they captive or wild? How did you know which bird was which? DId the birds have access to females? DId you observe every day during the specified time periods? Is this a total of all calls or a sample?

The origin of the birds and the location have been explained, they are wild birds so we think it is obvious that they had access to the females. As for the date of observation, we stated it in the methodology. We wrote that on randomly selected days.

  1. How did you do this?

Based on sound analysis software.

  1. What type of pairwise comparison did you use here?

Multiple pairwise comparison of mean ranks what we explained in the methodology.

  1. P=0.000 is not statistically correct. Please rephrase to p<0.001.

We improved.

Figure 2. 0.14s

We improved.

  1. Is this just random variation or is this a trend?

It is difficult for us to answer this question, it requires further research.

Table 2. I am not clear that you have mentioned chi squared in your methods?

Many publications give the value of chi2 but we can always delete it if necessary.

Table 2. If you look at the data range, there is almost perfect overlap between the IV and V data. This suggests that there shouldn't be a significant p value. Please provide your data outputs for reference

It is difficult for us to relate to this, because the statistical program showed statistical differences, because the data had a distribution deviating from the normal distribution.

  1. wording doesn’t make sense

We have removed this sentence.

Table. 4 This is all very nice - but it isn't clear why this is needed.

If necessary, we can always delete it.

226.cocks

We improved.

  1. This needs to be mentioned previously.

We improved.

  1. Why does this matter and what is the biological relevance?

In accordance with this suggestion, we have highlighted a relevant note in the conclusions.

  1. This is a weak last sentence for the conclusion. Bring in details as to why this research matters.

In accordance with this suggestion, we have highlighted a relevant note in the conclusions.

  1. Please include the publication location here

It is a publishing (book) item and it is given in such a format in scientific publications

  1. Italicise here

We improved.

  1. Include publication location.

It is a publishing (book) item and it is given in such a format in scientific publications.

Round 2

Reviewer 3 Report

Dear Authors,

Thank you for submitting a revised version of this paper. The methods are still very unclear and the background (and implications of the study) are still poorly provided. Please refer to the original feedback and ensure that all points have been sufficiently addressed.

Author Response

We are sending additional explanations

  1. There needs to be some background here - why does the study matter?

We added two sentences regarding the background in Simple summary “Research on bird vocalizations are an useful tool for researches on behavior. Singing behavior to reproductive behaviour such as can affect territory formation and mate choice.”

  1. What are the implications of this study?

We added few sentences regarding the background in Introduction:

“Research on avian vocalizations provides an useful tool for monitoring populations and researches on behavior. It relatively easy to detect and characterize song structure and components and to relate singing behavior to reproductive behaviour such as can affect territory formation and mate choice. To date, there have been few studies on the subject of pheasant vocalization. The results of the present study conducted in urban areas can be the basis for a comparison with populations of pheasants inhabiting agricultural areas”.

Table 1. This is unclear. Where were the birds? Are they captive or wild? How did you know which bird was which? DId the birds have access to females? DId you observe every day during the specified time periods? Is this a total of all calls or a sample?

We added to the methodology:

“The pheasants studied were birds living in the wild in the city park. We recorded the voices of individual roosters separately for each individual, which was preceded by observations in the field. The males had access to the hens. The choice of days on which recordings were made was irregular and depended on the activity of the recorded roosters. The number of recordings for analysis represents a random sample of all noises.”

  1. How did you do this?

We added to the methodology:

“This was done by analyzing the sonograms of individual pheasant vocalisations using a sound analysis program”.
